# Interaction between *Dickeya dianthicola* and *Pectobacterium parmentieri* in Potato Infection under Field Conditions

**DOI:** 10.3390/microorganisms9020316

**Published:** 2021-02-04

**Authors:** Tongling Ge, Fatemeh Ekbataniamiri, Steven B. Johnson, Robert P. Larkin, Jianjun Hao

**Affiliations:** 1School of Food and Agriculture, University of Maine, Orono, ME 04469, USA; tongling.ge@maine.edu (T.G.); fatemeh.ekbataniamiri@maine.edu (F.E.); 2Cooperative Extension, University of Maine, Orono, ME 04469, USA; stevenj@maine.edu; 3New England Plant, Soil, and Water Lab, USDA-ARS, University of Maine, Orono, ME 04469, USA; bob.larkin@usda.gov

**Keywords:** vacuum infiltration, blackleg and soft rot, synergy

## Abstract

*Dickeya* and *Pectobacterium* spp. both cause blackleg and soft rot of potato, which can be a yield-reducing factor to potato production. The purpose of this study was to examine the interaction between these two bacterial genera causing potato infection, and subsequent disease development and yield responses under field conditions. Analysis of 883 potato samples collected in Northeastern USA using polymerase chain reaction determined that *Dickeya dianthicola* and *P. parmentieri* were found in 38.1% and 53.3% of all samples, respectively, and that 20.6% of samples contained both *D. dianthicola* and *P. parmentieri*. To further investigate the relationship between the two bacterial species and their interaction, field trials were established. Potato seed pieces of “Russet Burbank”, “Lamoka”, and “Atlantic” were inoculated with bacterial suspension of *D. dianthicola* at 10^7^ colony-forming unite (CFU)/mL using a vacuum infiltration method, air dried, and then planted in the field. Two-year results showed that there was a high correlation (*p* < 0.01) between yield loss and percent of inoculated seed pieces. In a secondary field trial conducted in 2018 and 2019, seed pieces of potato “Shepody”, “Lamoka” and “Atlantic” were inoculated with *D. dianthicola*, *P. parmentieri*, or mixture of both species, and then planted. In 2019, disease severity index, as measured by the most sensitive variety “Lamoka”, was 16.2 with *D. dianthicola* inoculation, 10.4 with *P. parmentieri*, 25.4 with inoculation with both bacteria. Two-year data had a similar trend. Thus, *D. dianthicola* was more virulent than *P*. *parmentieri*, but the co-inoculation of the two species resulted in increased disease severity compared to single-species inoculation with either pathogen.

## 1. Introduction

Blackleg and soft rot (BSR) of potato (*Solanum tuberosum*) is caused by many bacterial species in the genera *Dickeya* and *Pectobacterium*. The predominant species of bacteria responsible for the disease varies depending on geographical locations. For example, *Pectobacterium atrosepticum* was dominant in Europe before 1970, but *Dickeya dianthicola* followed by *D. solani*, have become dominant in some European countries in recent decades [1,2]. An outbreak of BSR in Northeastern USA was caused by *D. dianthicola* in 2015 and the following years, while *D. chrysanthemi* was found in central USA, and *Pectobacterium* spp. was found to be widespread in the USA [3,4,5,6,7,8,9]. Symptoms of BSR were expressed as rot and blackened stems (blackleg), wilting plants and decayed tubers (soft rot), which have threatened potato production, resulting in millions of dollars in losses for the potato industry [3].

BSR caused by a pathogen complex adds an additional layer of difficulty in the development of effective control measures. One plant may be infected by more than one pathogen [10,11]. Such a complex disease structure is determined based on the interactions between pathogen species, host plant, biotic and abiotic environmental factors [10,12,13,14,15].

It is now generally believed that *D. dianthicola* is primarily a seed-borne pathogen, and most *Pectobacterium* spp. are soil inhabitants but can be seed-borne as well. Therefore, stored tubers are the most important source in harboring bacterial pathogens and initiating disease. Pathogens surviving in soil can colonize roots, then penetrate and move into the xylem [16]. Some studies indicate that *P. parmentieri* is less virulent than *P. carotovorum*, owing to the lack of Type III secretion system (T3SS) [17,18]. However, we have demonstrated that *P. parmentieri* isolates are highly pathogenic on both the tubers and stems of potato [19].

In the USA outbreak of BSR in 2015 and the following years, *Dickeya dianthicola* was determined to be the predominant pathogen involved, based on its primary and predominant detection and isolation from field samples. However, the detection frequency of this species progressively declined in the following years. In contrast, some *Pectobacterium* spp. were found on postharvest tubers and later identified in symptomatic potato plants in increasing amounts in subsequent years. Common species of BSR causal agents included *P. parmentieri*, *P. polaris*, *P. carotovorum* subsp. *carotovorum*, *P. c.* subsp. *oderiferium*, and other *Pectobacterium* spp. [5,7,19,20]. The above species were classified as one species, *Erwinia carotovorum*, before 1981, which were further differentiated into several subspecies soon afterwards [21,22,23]. In addition, *P. parmentieri* extended its distribution beyond the Northeastern USA, causing blackleg and soft rot of potato [24].

Whether the co-existence of *Dickeya* and *Pectobacterium* spp. has advantages for pathogen infection is a key subject that impacts both biological study and disease management. Our questions were: Is the predominance of bacterial species influenced by environmental factors or bacterial virulence? Is there an enhanced response when both species co-occur in the same infected plant tissue? The aims of this study were to understand how bacterial infection affected potato growth and the corresponding yield response. Specifically, we wanted to determine how multiple species of pathogens interact, and how they affect disease development and yield response.

## 2. Materials and Methods

### 2.1. Frequency and Distribution of Dickeya dianthicola and Pectobacterium parmentieri in Naturally Infected Potato

From 2015 to 2020, symptomatic potato stems and tubers showing BSR were collected from fields in most of the Northeastern state by growers, consultants, or vegetable pathologists. All samples were processed and assayed at the University of Maine. Segments of diseased stem tissue or tuber peel were obtained from the samples. Bacteria were extracted from the tissue sap by incubating them in sterile distilled water [25]. Genomic DNA was extracted from the stem- or tuber-derived sap using the FastDNA^®^ SPIN Kit (MP Biomedical, Santa Ana, CA, USA). Concentration of DNA was estimated using NanoDrop 2000c spectrophotometer (Thermo Fisher Scientific Inc, Wilmington, DE, USA), and diluted to 10 ng/µL for polymerase chain reaction (PCR). A total of 883 DNA samples were selected and amplified by PCR with primer pairs of PW7011 specific for *P. parmentieri*, and DIA-A specific for *D. dianthicola* [26,27]. PCR amplifications were performed in a 25 µL total volume reaction system containing 17.9 µL of sterile water, 5 µL of 5× PCR buffer, with MgCl_2_ at a final concentration of 0.25 mM dNTPs, 0.1 µM of each pair of primers, 0.5 U GoTaq DNA Polymerase and 1 µL of DNA. All PCR reagents were from Promega Corporation (Madison, MI, USA). The thermal cycler for primer DIA-A was programmed for an initial denaturation of 5 min at 94 °C, followed by 35 cycles of 1 min at 94 °C, 30 s at 53 °C and 1 min at 72 °C, with a final elongation of 10 min at 72 °C, while the annealing temperature of primer PW7011 was set at 67 °C for 30 s.

### 2.2. Tuber Inoculation Using Vacuum Infiltration

*Dickeya dianthicola* strain ME30 and *P. parmentieri* strain ME175 were cultured on crystal violate pectate (CVP) plates at 28 °C for 48 h. The culture was transferred to tryptic soy broth (TSB) and incubated on a shaker overnight at 28 °C and 120 rpm. The chamber was disinfected with 10% bleach between treatments. At plant inoculation, the prepared bacterial culture was diluted with non-autoclaved tap water and adjusted to 10^7^ CFU/mL as working inoculum. Well-suberized seed pieces were washed and submerged into the bacteria suspension in a vacuum chamber (BVV10GL, BEST VALUE VACS, Naperville, IL, USA) and tightly sealed. The chamber held between −60 and −80 kPa for 15 min [28,29]. A control was set up by treating seed pieces with tap water (pathogen free) as “non-inoculated” through the same procedure in the vacuum chamber. The treated tubers were air dried and planted one week after inoculation.

### 2.3. Effects of Bacterial Inoculation on Potato Growth

Field studies were conducted in 2019 and 2020 at Presque Isle, Maine. A randomized complete random block design (RCBD) was used with four replicate blocks. Potato varieties “Lamoka”, “Atlantic”, and “Russet Burbank” were inoculated with *D. dianthicola* strain ME30 using the vacuum infiltration method as described above. For each variety, 50 tubers were planted on 23 May 2019 and 27 May 2020, and each treatment contained a mixture of bacteria-inoculated and bacteria-free tubers at different percentages, including 0, 10%, 20%, 40% and 80%, which were arranged in RCBD. Fertilizer (N:P:K = 14:14:14) was applied at time of planting at a rate of 12.3 kg/a. Nuprid 2SC (a.i. 21.4% imidacloprid) was also applied at time of planting at 14.6 mL/a to control insects. Plant stand was evaluated on 1 July 2019 and 13 July 2020. Plants were visually inspected for the development of BSR symptoms. Disease severity was evaluated weekly or every other week after symptoms appeared and rated from 0 to 5, with 0 being healthy and 5 being dead plants. Potato vines were killed with two applications of Reglone at 17.5 mL/a. Potatoes were harvested on 5 September 2019, and 2 October 2020. Total yield was measured.

### 2.4. Interaction between D. dianthicola and P. parmentieri in Potato Infection

Field studies were performed in 2018 and repeated in 2019 in Presque Isle, Maine. *Dickeya dianthicola* strain ME30 and *P. parmentieri* strain ME175 were used. Seed pieces of potato “Lamoka”, “Atlantic”, and “Shepody” were inoculated with either ME30, ME175, or mixture of both bacterial species using the vacuum infiltration method as described above. For the co-inoculation, cell suspension of strains ME30 and ME175 were mixed together at a 1:1 ratio with the final concentration of 10^7^ CFU/mL, which was then used for vacuum infiltration. The treated tubers were air dried for at least one week before planting. A randomized complete block design (RCBD) was applied with four replications in the field. Thirty seed pieces were planted in each plot on 25 May 2018 and 28 May 2019. Field management was as described in the above trial. Emergence was assessed as the number of emerged plants per plot on 2 July 2018 and 1 July 2019. Plants were visually inspected for the development of blackleg symptoms with 0 to 5 disease severity scale, as described above. Disease incidence was calculated as DSD = number of symptomatic stems/total number of stems. Disease severity index (DSI) was calculated as DSI = (sum (disease scale frequency × score of disease scale))/((total number of stems) × (maximal disease severity index)) × 100. Potatoes were harvested on 5 September 2018 and 19 September 2019. Total yield was measured.

### 2.5. Statistical Analysis

Statistical analysis was conducted using the SAS statistical program (SAS university edition, Red Hat (64-bit) version, SAS Institute Inc., Cary, NC, USA). Categorical variables were analyzed using chi-square test. Numerical variables, including emergence, disease incidence, disease severity index and yield were analyzed using the ANOVA procedure with Tukey’s multiple range test at a significance level α = 0.05. Emergence loss and yield loss (*Y*) were regressed against percentage of seed infection (*X*) using linear regression equation *Y* = b_0_ + b_1_*X*. Emergence loss was expressed as (noninfected plot stand count − infected plot stand count)/noninfected plot stand count. Yield loss = (yield in noninfected plot − yield in infected plot/yield in noninfected plot).

## 3. Results

### 3.1. Frequency and Distribution of Dickeya dianthicola and Pectobacterium parmentieri in Naturally Infected Potato

Potato samples collected from fields were assayed using PCR to detect both *Dickeya* and *Pectobacterium* spp. Out of the 883 samples, 761 were from potato stems, and 122 were from potato tubers. Primers DIA-A, specific to *D. dianthicola*, and PW7011 specific to *P. parmentieri* were used in the detection. Through the six years, 38.1% of samples contained *D. dianthicola* and 53.3% of samples contained *P. parmentieri*. Furthermore, 20.6% of samples contained both *D. dianthicola* and *P. parmentieri*, and 29.2% of samples contained neither *D. dianthicola* nor *P, parmentieri* (Table 1). Chi-square test showed that the percentage of *D. dianthicola* in stems (40.3%) was significantly (*p* < 0.001) higher than in tubers (23.8%), whereas the percentage of *P. parmentieri* positive on stems (51.8%) was significantly (*p* < 0.01) lower than the one on tubers (63.1%) (Table 2).

There was a trend that the percentage of samples infected by *P. parmentieri* increased during the years, from 30.4% in 2015 to 95.3% in 2020 (Table 1). On the contrary, the percentage of total samples infected by *D. dianthicola* declined from 41.7% to 9.5% through the six years (Table 1).

### 3.2. Effects of Bacterial Inoculation on Potato Growth

An extremely dry growing season in 2020 resulted in low emergence, with the non-inoculated treatment having an emergence rate of 33%, 56% and 67% for “Lamoka”, “Atlantic”, and “Russet Burbank”, respectively (Table 3). Potato varieties showed different responses in yield and emergence to bacterial inoculation (Table 3). The responses were evaluated with regression models. Emergence loss and inoculation had a linear relationship with correlation coefficients of 0.66 on “Lamoka”, 0.79 on “Atlantic” and 0.60 on “Russet Burbank”, while yield loss and inoculation on “Lamoka” and “Atlantic” showed linear relationships with correlation coefficients of 0.66 and 0.74, respectively (Figure 1. The level of disease susceptibility was estimated by the disease incidence and the slope of the regression equation between emergence loss and inoculation. From high to low, disease incidence in 2019 was observed on “Lamoka” (31.7%), “Atlantic” (4.4%), and “Russet Burbank” (0) (Figure 2). Slope of the regression equation between emergence loss and inoculation was “Lamoka” (0.687), “Atlantic” (0.475), and “Russet Burbank” (0.332) (Figure 1). “Lamoka” was the most susceptible variety, with 100% inoculated seed resulted in 68.7% emergence loss and a corresponding 68.1% yield loss (Figure 1). For the tolerant variety “Russet Burbank”, yield was not significantly affected by bacterial inoculation because emergence was not significantly reduced (Figure 1).

### 3.3. Interaction between D. dianthicola and P. parmentieri in Potato Infection

Three potato varieties had different responses when inoculated with either *D. dianthicola P. parmentieri* or mixture of both bacterial species, with “Lamoka” being the most susceptible variety, “Atlantic” and “Shepody” being the less susceptible varieties. This result was consistent in both years. For example, in 2019, disease symptoms with inoculation of the bacterial mixture appeared on “Lamoka” at 49 days post planting with the disease severity index of 5.7, while the disease severity indexes of “Atlantic” and “Shepody” at 49 days post planting were 0.75 and 1.6, respectively (Figure 3). At the end of the growing season, the disease severity index of “Lamoka” was 25.4, while “Atlantic” and “Shepody” showed a relatively lower disease severity index of 7.0 and 7.1, respectively (Figure 3).

*Dickeya dianthicola* was more aggressive than *P. parmentieri* in the field. For example, in the season of 2019 and on the most susceptible variety “Lamoka”, symptoms were first observed on potato plants inoculation with *D*. *dianthicola* than *P. parmentieri.* Disease severity index was 6.7 by *D*. *dianthicola* inoculation but 0.7 by *P. parmentieri* inoculation (Figure 3). Furthermore, by the end of the season, the disease severity index of “Lamoka” caused by *D. dianthicola* was 16.2 higher than that caused by *P. parmentieri* (10.4) (Figure 3). However, the mixture of both bacteria showed the highest disease level, with disease severity index being 25.4 (Figure 3). Thus, the mixture of two bacterial species was aggravated in plant infection compared to either of the two species. The responses of “Shepody” and “Atlantic” were similar to “Lamoka” with infection by either a single or a mixture of bacterial species, although the difference among treatments was not significant (*p* > 0.05), because both varieties were less susceptible to blackleg (Figure 3). At early growing stage of potato, a higher disease incidence was observed with *D. dianthicola* inoculation compared to the treatment with *P. parmentieri* inoculation. Although inoculation with the mixture of bacterial species initially showed disease level slightly lower than that of *D. dianthicola*, the disease severity index of the mixture inoculation soon surpassed *D. dianthicola* alone. Although not statistically significant, the disease severity index of treatments in 2018 showed a trend of *D. dianthicola* inoculation resulting in a numerically higher disease severity index than *P. parmentieri* inoculation, regardless of potato varieties (Figure 3).

All bacterial inoculations significantly reduced potato yield compared to the non-inoculated treatment in 2019; yield differences in the 2018 trial were not significant (Table 4). The treatment with a mixture of bacterial species did not show significant difference compared to only *D. dianthicola* inoculation (Table 3). Thus, *D. dianthicola* was a key to causing higher yield loss when potatoes were either infected solely with *D. dianthicola* or coinfected with *D. dianthicola* and *P. parmentieri*.

## 4. Discussion

*Dickeya dianthicola* was found to be the predominant species causing BSR in potato during the 2015 outbreak in the Northeastern USA. There was a trend where the number of samples received which were infected by *P. parmentieri* increased during subsequent years, but the percentage of total samples infected by *D. dianthicola* declined through the six years of the study. *Pectobacterium parmentieri* developed into a separate problem along with *D. dianthicola*, which was persistently detected as the predominant pathogen in subsequent years. Furthermore, *D. dianthicola* was more likely a pathogen associated with stems, and *P. parmentieri* was highly associated with potato tubers. It is possible that *P. parmentieri* is a better tuber-rotter than *D. dianthicola*.

We identified 20.6% infected potato plants contained both *D. dianthicola* and *P. parmentieri*. In addition, several other *Pectobacterium* spp. have also frequently been found in naturally infected seed potatoes (unpublished). It seemed the coexistence of multiple bacterial species is common in blackleg, forming a pathogen complex. Since *D. dianthicola* was consistently isolated as the predominant species, it is believed that *D. dianthicola* was the primary pathogen that aggressively caused disease in the field. Many *Pectobacterium* spp. can survive well in soil or they are soil inhabitants. Since they are weak pathogens, they may not cause infection. However, their availability in field allows them to follow the initiation of infection by *Dickeya* spp. and thus increase disease severity. However, come of *Pectobacterium* spp. such as *P. parmentieri* can be carried over to storage, where they seem to be highly active in causing tuber rot. A portion of the bacteria will move with seed pieces and may infect potato independently or collaboratively with *D. dianthicola*. From this point, BSR is a seedborne disease regardless of the pathogen taxon.

It is possible that *D. dianthicola*, compared to *P. parmentieri*, has more virulence-related factors in plant infection, such as enzymes to degrade pectin and cellulose [30], secretion systems to transfer virulence effectors [17,31]. Regarding pathogenicity, *D. dianthicola* might have advantages over *P. parmentieri* owing to the lack of T3SS in *P. parmentieri* [17,18], as T3SS contributes to the virulence of bacteria during the early stages of infection [32,33]. Studies have reported that *D. dadantii* exhibited a high number of pectinase-related enzymes that would enhance the degradation of pectin [34], and *Dickeya* spp. possess complex regulatory networks in order to express virulence genes [35]. However, detailed research about virulence-related mechanisms has not been established for *D. dianthicola*.

Our two-year data indicated that *D. dianthicola* causes a higher level of BSR severity than *P. parmentieri*, and inoculation with both bacterial species did increases BSR severity. This suggested that *D. dianthicola* should be considered the primary pathogen in disease control. Meanwhile, due to the wide distribution of *P. parmentieri*, it is possible that both bacterial species co-existed in the environment and co-infected or -damaged potato plants. This could be a consequence of synergy between the two bacterial species. In production, if *D. dianthicola* is eliminated, the less-aggressive *P. parmentieri* would likely have a lower impact on potato growth and yield. More importantly, it would remove the complex of co-infection with two or more bacteria. Therefore, it is necessary to further investigate how *Dickeya* spp. and *Pectobacterium* spp. interact with each other inside potato plants in increasing disease severity.

Competition, cooperation, and coexistence are major means of interactions of bacteria that co-exist in a same niche [36]. In this study, there was no antagonistic effect between *D. dianthicola* and *P. parmentieri* when they coexisted, but they aggravated the disease epidemic at a higher level. Surprisingly, the highest disease severity caused by the mixture of *D. dianthicola* and *P. parmentieri* did not transfer into the highest yield loss. We interpreted that yield is constrained by early infection and stand loss, and compensation by the potato plant, and less affected by late-season disease progress. Furthermore, among the three bacterial treatments, one with *P. parmentieri* had a lesser impact on yield compared to the other two. This implied that *D. dianthicola* was more aggressive or virulent than *P. parmentieri*.

Yield losses of potato caused by artificial inoculation of *D. dianthicola* were evaluated in this study. There was a high correlation of percentage of seed inoculation/contamination with lack of emergence, as well as with yield reduction. This was because yield per area of field was highly correlated with plant emergence or the total number of plants. BSR showed up in the growing season at a low level and most of the plants survived. Therefore, a low percentage of in-season disease did not significantly affect potato yield. In addition to disease severity, yield is also dependent on weather conditions, such as precipitation and soil temperature, soil properties, pathogen aggressiveness, and the ability of the crop to compensate for stand loss. Based on our results, “Lamoka” was the most susceptible variety to BSR, while “Russet Burbank” was more tolerant, “Atlantic” and “Shepody” were moderate. Such information may be incorporated in breeding program for BRS resistance.

In conclusion, tuber contamination with either *D. dianthicola* or *P. parmentieri* caused direct stand loss that transferred to yield loss, and the percent of tuber infection was negatively correlated with emergence when a single species of pathogen-infected potato, *D. dianthicola*, caused more symptoms than *P. parmentieri*. However, co-infection of the two species showed even higher disease severity of BSR, and a synergistic effect of bacterial species in plant infection.

## Figures and Tables

**Figure 1 microorganisms-09-00316-f001:**
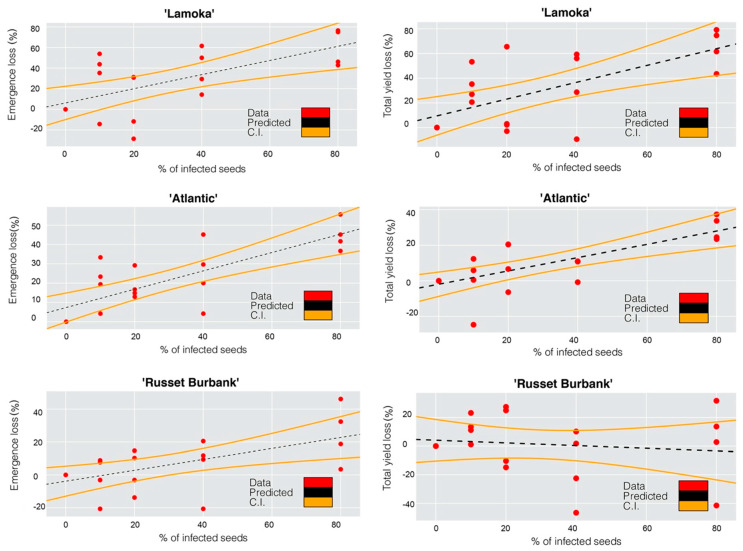
Correlation between (1) (**left panels**) emergence loss and infection percent of potato “Lamoka” (*r* = 0.66, *p* = 0.0017), “Atlantic” (*r* = 0.79, *p* < 0.0001), “Russet Burbank” (*r* = 0.60, *p* = 0.0056) seed pieces inoculated with *Dickeya dianthicola*; (2) (**right panels**) yield loss and infection percent of potato “Lamoka” (*r* = 0.66, *p* = 0.0014), “Atlantic” (*r* = 0.74, *p* = 0.0002), “Russet Burbank” (*r* = 0.13, *p* = 0.5850) seed pieces inoculated with *Dickeya dianthicola*. Predicted data and confident intervals (CI) were shown as the dash lines and orange curves, respectively, 2020.

**Figure 2 microorganisms-09-00316-f002:**
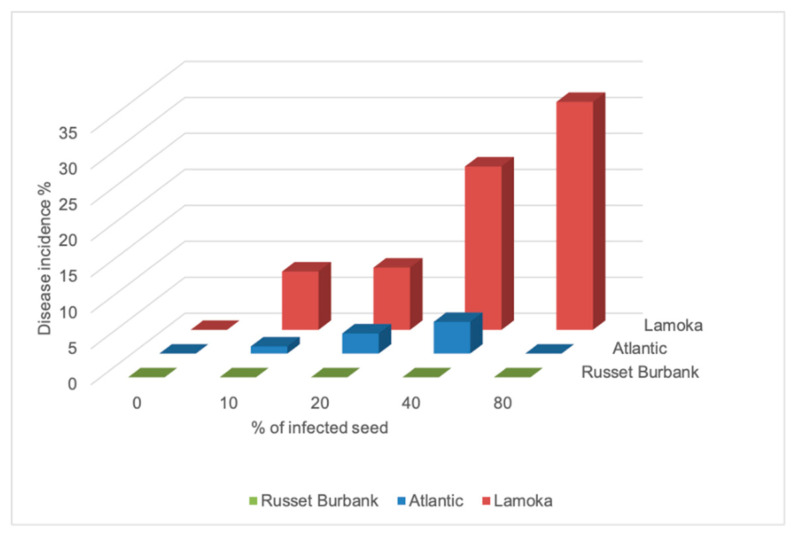
Disease incidence affected by infection of potato “Lamoka”, “Russet Burbank”, and “Atlantic” seed pieces inoculated with *Dickeya dianthicola* in 2019.

**Figure 3 microorganisms-09-00316-f003:**
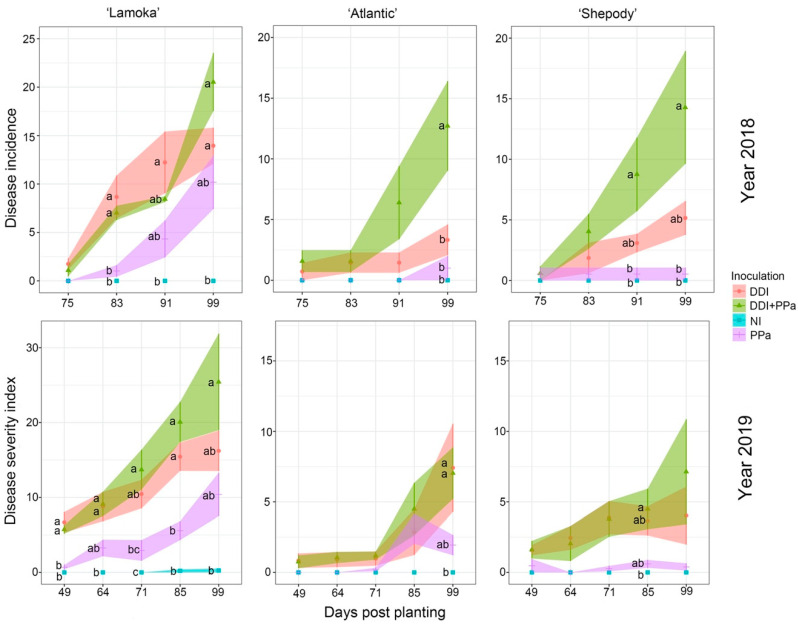
Disease progress of blackleg in potato “Lamoka”, “Atlantic” and “Shepody” grown from seed pieces inoculated with either *Dickeya dianthicola* (DDi), *Pectobacterium parmentieri* (PPa), or DDi + PPa under field conditions in 2018 (**upper panels**) and 2019 (**bottom panels**). Non-inoculated (NI) plants were used for control. Disease incidence or disease severity index was analyzed using the ANOVA procedure with Tukey’s multiple range test at a significance level α = 0.05. Mean values at same day point with different letters were significantly different (*p* < 0.05).

**Table 1 microorganisms-09-00316-t001:** Detection of *Dickeya* and *Pectobacterium* spp. in symptomatic potato stems or tubers using polymerase chain reaction (PCR).

Year	Number of Samples	*D. dianthicola* (%)	*P. parmentieri* (%)	*D. dianthicola* + *P. parmentieri* (%)	Both Negative
2015	319	133 (41.7)	97 (30.4)	39 (12.2)	128 (40.1)
2016	230	114 (49.6)	134 (58.3)	72 (31.3)	54 (23.5)
2017	171	60 (35.1)	132 (77.2)	50 (29.2)	29 (17.0)
2018	81	14 (17.3)	45 (55.6)	9 (11.1)	31 (38.3)
2019	61	13 (21.3)	43 (70.5)	10 (16.4)	15 (24.6)
2020	21	2 (9.5)	20 (95.3)	2 (9.5)	1 (4.8)
Total	883	336 (38.1)	471 (53.3)	182 (20.6)	258 (29.2)

**Table 2 microorganisms-09-00316-t002:** Chi-square test on tissue distribution of *Dickeya dianthicola* (DDi) and *Pectobacterium parmentieri* (PPa) on potato tissues showing blackleg and soft rot symptoms.

	Number (Percentage) of Sample					
	Stem (Observed)	Tuber (Observed)	Tuber (Expected) ^a^	Chi-Sq	df	*p* Value
DDi positive	307 (40.3%)	29 (23.8%)	49 (40.3%)	13.64	1	<0.001
DDi negative	454 (59.7%)	93 (76.2%)	73 (59.7%)			
Total	761	122	122			
PPa positive	394 (51.8%)	77 (63.1%)	63 (51.8%)	6.43	1	<0.01
PPa negative	367 (48.2%)	45 (36.9%)	59 (48.2%)			
Total	761	122	122			

^a^ The number of tuber samples showing pathogen (DDi/PPa) positive or negative with the assumption that there was no association between pathogen and tissue.

**Table 3 microorganisms-09-00316-t003:** Effects of infection level of seed pieces with *Dickeya dianthicola* on yield and emergence of potato in 2020.

Variety	Infected Seeds (%)	Total Yield (kg/a)	Emergence (%)	Total Yield/Plant (kg)
Lamoka	0	126.5	33.0	1.19
	10	86.73	20.0	1.17
	10	97.14	28.6	1.06
	40	80.04	18.0	1.23
	80	43.52	13.0	1.10
Atlantic	0	186.1	56.0	0.923
	10	189.8	44.6	1.18
	20	164.5	46.0	1.00
	40	171.7	41.6	1.18
	80	130.8	31.0	1.17
Russet Burbank	0	215.0	67.0	0.903
	10	187.0	67.6	0.777
	20	194.2	65.0	0.831
	40	239.7	62.6	1.08
	80	204.3	49.0	1.16

**Table 4 microorganisms-09-00316-t004:** Effects of seed tuber inoculation with Dickeya dianthicola (DDi), Pectobacterium parmentieri (PPa) and non-inoculated (NI) on emergence and yield of potato varieties in years 2018 and 2019.

Variety	Inoculation	2018		2019	
		Emergence (%)	Yield (kg/a)	Emergence (%)	Yield (kg/a)
Lamoka	NI	95.0	148.5	98.3	208.6 a
	DDi	95.0	126.0	85.0	151.6 b
	PPa	93.3	148.0	90.8	191.7 b
	DDi and PPa	96.7	154.7	89.2	151.1 b
Atlantic	NI	96.7	162.9	85.8	185.5 a
	DDi	86.7	156.2	76.7	142.6 b
	PPa	83.3	165.5	84.2	152.7 b
	DDi and PPa	86.7	122.8	82.5	153.2 b
Shepody	NI	96.7	159.4	95.0	237.0 a
	DDi	96.7	127.3	86.7	174.0 b
	PPa	96.7	198.1	87.5	183.2 b
	DDi and PPa	91.7	139.6	90.8	171.5 b

Mean values followed by different letters were significantly different (*p* < 0.05).

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
