# Peer review of "Interaction between Dickeya dianthicola and Pectobacterium parmentieri in Potato Infection under Field Conditions"

_microorganisms, 2021, doi:10.3390/microorganisms9020316_

Round 1
Reviewer 1 Report
The manuscript is well written as a plant pathology paper which, in my opinion, will be best aligned with the scope of the journal Agriculture (MDPI) rather than the current journal (Microorganisms). The authors report an interaction between Dickeya dianthicola and Pectobacterium parmentieri using potato as the test plants, but the nature of this interaction is yet to be elucidated. It is clear that combining the two pathogens resulted to higher disease severity index, suggesting an additivity effect when both bacterial genera are present. The crucial question here is what is the nature of this additivity phenomenon? Also, it is unclear why the authors used cv. Russet Burbank in one study (Fig. 2) but used cv shepody in the disease progression study (Fig. 3). Including a more tolerant variety such as Russet BurBank instead of using two moderate cultivars could have provided another angle to the story.
The authors should consider using the International System of Units (SI) throughout the manuscript.
Minor comments:
Line 88: What is the amount of DNA in 1 microlitre of DNA added to the PCR reaction.
Line 98: Was the tap water autoclaved?
Reviewer 2 Report
The manuscript of Ge and co-workers targets the co-presence of plant pathogenic Dickeya dianthicola and Pectobacterium parmentieri (they both belong to so-called Soft Rot Pectobacteriaceae – SRP) both in naturally infected and in artificially inoculated and field-planted potato tubers. Both SRP species are plant pathogens causing high and increasing losses in agricultural crops worldwide in many regions including USA. The topic of the manuscript is interesting not only for growers/farmers but also for scientists working on these bacteria as studies about co-presence/co-infections with SRP bacteria are rather rare. The authors have shown that the both bacterial species not only co-exist in the same crop but also when together they are more virulent causing higher losses. These observations have been made under field conditions and in two growing seasons – such results are overall very important for our better understanding of the infection process. This is a very solid, partially descriptive, partially mechanistic work done with care and very scientifically sounded. I do not have any comments about preparations of the experiments and their analyses, as well the obtained results are in depth discussed by the authors. The manuscript should be accepted as it is.
Author Response
We thank the reviewer for supporting this study.